# Demographic and health community-based surveys to inform a malaria elimination project in Magude district, southern Mozambique

Beatriz Galatas [1,2] Ariel Nhacolo [1] Helena Marti,[1,2] Humberto Munguambe,[1] Edgar Jamise,[1] Caterina Guinovart,[2] Laia Cirera,[1,2] Felimone Amone,[1] Eusebio Macete,[1,3] Quique Bassat,[1,2,4,5] Regina Rabinovich,[2,6] Pedro Alonso,[1,2] Pedro Aide,[1,7] Francisco Saute,[1] Charfudin Sacoor[1]

BG and AN contributed equally.

For numbered affiliations see end of article.

**Correspondence to**
Dr Beatriz Galatas;
beatriz.galatas@isglobal.org

## ABSTRACT

**Objectives** A Demographic and Health Platform was established in Magude in 2015, prior to the deployment of a project aiming to evaluate the feasibility of malaria elimination in southern Mozambique, named the Magude project. This platform aimed to inform the design, implementation and evaluation of the Magude project, through the identification of households and population; and the collection of demographic, health and malaria information.

**Setting** Magude is a rural district of southern Mozambique which borders South Africa. It has nine peripheral health facilities and one referral health centre with an inpatient ward.

**Intervention** A baseline census enumerated and geolocated all the households, and their resident and non-resident members, collecting demographic and socio-economic information, and data on the coverage and usage of malaria control tools. Inpatient and outpatient data during the 5 years (2010 to 2014) before the survey were obtained from the district health authorities. The demographic platform was updated in 2016.

**Results** The baseline census conducted in 2015 reported 48 448 (92.1%) residents and 4133 (7.9%) non-residents, and 10 965 households. Magude's population is predominantly young, half of the population has no formal education and the main economic activities are agriculture and fishing. Houses are mainly built with traditional non-durable materials and have poor sanitation facilities. Between 2010 and 2014, malaria was the most common cause of all-age inpatient discharges (representing 20% to 40% of all discharges), followed by HIV (12% to 22%) and anaemia (12% to 15%). In early 2015, all-age bed-net usage was between 21.8% and 27.1% and the reported coverage of indoor residual spraying varied across the district between 30.7% and 79%.

**Conclusion** This study revealed that Magude has limited socio-economic conditions, poor access to healthcare services and low coverage of malaria vector control interventions. Thus, Magude represented an area where it is most pressing to demonstrate the feasibility of malaria elimination.

**Trial registration number** NCT02914145; Pre-results.

## Strengths and limitations of this study

► A Demographic and Health Platform (DHP) was established in Magude in 2015 to enumerate and geolocate all the households, and their resident and non-resident members in order to update the demographic information of the district which was last collected in 2007 during a national census.

► Second census round was conducted one year later acknowledging that detailed population data collected at individual level at one point in time may miss individuals or households that can be captured after a later update.

► The census rounds were planned in close collaboration with the community leaders and district authorities, and counted with intensified training of fieldworkers, and a strong component of field and data supervision by experienced demographers.

► Data collected through the DHP could have been affected by inaccuracies during data collection or entry, or by recall or desirability bias of census participants.

► Inpatient and malaria outpatient information was limited by the quality and accuracy of the data at the time it was collected, by disease-specific interventions or changes in diagnostics, referral or reporting practices.

## BACKGROUND

Mozambique is one of the countries with the highest malaria burden in the world.[1] Malaria prevalence among children of 6 to 59 months of age is heterogeneous within the country ranging from high transmission intensity in the North (>50%) to less than 3% in the South, as reported in the latest malaria survey conducted in 2015.[2] Lower prevalence in the southern region may be related to the great progress against malaria in the past decades, partly as a result of the regional initiatives aiming for malaria elimination in the area.

In line with the vision of a malaria free world established by the WHO in its Global Technical Strategy for 2016 to 2030,[3] the National Malaria Control Programme of Mozambique (NMCP) decided to redefine its strategic objectives in order to include the implementation of malaria elimination activities in the South. In this context, a malaria elimination project named the Magude project was designed and evaluated in Magude district, Maputo province, by the Manhiça Health Research Centre (CISM) and the Barcelona Institute of Global Health (ISGlobal), to assist the NMCP in adopting a malaria elimination strategy based on local evidence.[4]

Prior to the initiation of the Magude project in 2015, there was limited and outdated information with regards to the number of individuals living in Magude, as well as to their demographic and socio-economic characteristics.[5] Thus, detailed information from the whole district was deemed crucial to inform the elimination strategies that had been planned for the following years in the district. The process of filling in this knowledge gap also aimed to identify and contact key leaders at provincial, district and at community level, to inform and engage them in the activities prior to their deployment. In this context, a Demographic and Health Platform (DHP) adapted from the health and demographic surveillance system (HDSS) method was established in the district of Magude in February of 2015 with the objective of providing reliable and updated demographic data to inform the project. This platform allowed to plan the activities and to provide a sampling frame to measure indicators in the community such as malaria prevalence, and to estimate the coverage of indoor residual spraying (IRS), coverage and usage of long-lasting insecticide treated nets (LLIN) and mass drug administration (MDA) campaigns. The DHP's permanent identification numbers were used to track individual's participation in each intervention of the Magude project longitudinally, thus allowing to identify and quantify potential challenges to the project, such as reasons for non-participation, or probable sources of imported infections. Data were also used to accurately

measure prevalence at the community stratified by age groups and place of residence. Overall, these findings were crucial to the design of the Magude project, and offered robust evidence to guide malaria elimination strategies in southern Mozambique.

This article presents the demographic, socio-economic and health characteristics of the population of Magude, as well as the coverage of malaria control interventions estimated through the baseline census conducted between February 2015 and June 2015. It also presents the burden of disease in Magude during the 5 years (2010 to 2014) before the DHP, using the inpatient and outpatient data obtained from the district health authorities. A summary of the demographic profile of Magude after updating the census between August 2016 and September 2016 is also provided

## METHODS
### Study area

The district of Magude is located in the North-Western part of Maputo province and borders with the districts of Massingir, Chókwe and Bilene, from Gaza province on the North and North-East; with the districts of Manhiça and Moamba, of Maputo province in the East and South and with the South African National Kruger Park, in the West (figure 1A).

Magude was selected as an appropriate demonstration area for the malaria elimination project as it was expected to pose the types of challenges that the NMCP would face when implementing a malaria elimination campaign in the south of the country. This is, there were more than 13 000 malaria cases reported in the district in 2014, with the majority of cases observed between January and May, suggesting that the epidemiology of malaria in the district was representative of most endemic areas in the country with the typical seasonal pattern coinciding with the rainy season.[6 7] Additionally, the socio-economic and infrastructural limitations reported for Magude made it sufficiently representative of a rural district of Mozambique, while still

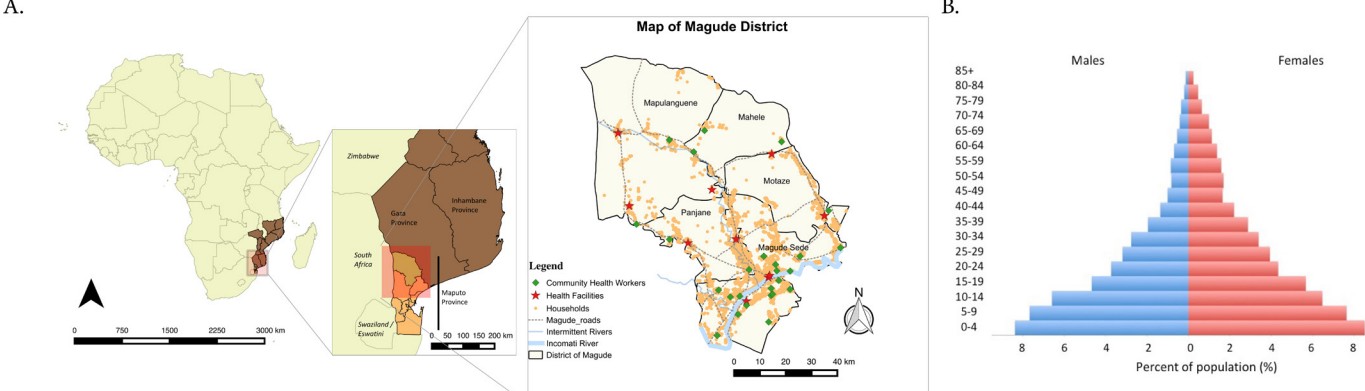

**Figure 1** Map of Magude district (2015). (A) Administrative and permanent river shape files were obtained from DIVA-GIS ((http://www.diva-gis.org/Data), and confirmed with Magude's key informants. GPS positions of households, health facilities and community health workers obtained directly from the field and mapped using QGIS. (B) Population pyramid of Magude district (2015) showing the proportion of 5-year age and sex groups out of the total population. GPS, geographic positioning system.

at reach of CISM's facilities (located in Manhiça district), which facilitated the logistics, supervision and quality control processes. Finally, the population of Magude had had very limited exposure to research projects or targeted innovative malaria interventions prior to 2015, having only received the programmatic IRS, LLINs and child immunisation campaigns conducted by the government. This allowed facing the challenges of working in an unexposed population that was not biassed by previous activities.

### Study design

A DHP was established in Magude in February 2015 by CISM, which was adapted from the HDSS previously established in Manhiça district also by CISM.[8] A baseline census was conducted at that time (February to June) to identify and enumerate all neighbourhoods, households and resident and non-resident individuals living in these households. Geographic positioning system coordinates were also captured for every household. All individuals were assigned a permanent and unique identification number, linked to the household number where they were first enumerated. A second census round was conducted between August 2016 and September 2016, to review and update all information collected during the baseline census and record live births, deaths and migrations that occurred between the two censuses. New households and new members were enumerated, and their information was collected to update the baseline databases.

This DHP defined a household as a structure or set of constructions where an individual or group of individuals live and share domestic activities and costs (such as eating and sleeping) and recognise one of them as their superior or chief, regardless of their kinship ties. Individuals were defined as residents if they had lived and slept in a household within the study area for a period of 3 months or more or intended to do so. Non-resident members of a household were defined as those who left the district or who had never lived in the study area for a period of three or more months, but who kept their roles in the households as head of household, husband or other important breadwinners, and paid regular visits to their households in Magude. These individuals leave their households for specific reasons such as work, studies or imprisonment; and would otherwise reside in the referred household. This category excluded offspring or other members who had left the household to live in their own households outside the study area, irrespective of whether they kept visiting the reference household or not.

### Data collection procedures

Baseline and updated data from the census rounds were collected through standardised questionnaires using Open Data Kit (https://opendatakit.org/) installed in Android tablets. Data were sent to a secure server at CISM using Wi-Fi. Information collected on households' socioeconomic characteristics included building materials used for the main house, source of water and electricity

and household assets and livestock. At the individual level, the information collected included the name, sex, date of birth, relation to the head of household, education and occupation of all residents and non-residents of Magude. Information was also collected on malaria prevention tools, including the possession and use of LLINs, and whether the house had received IRS in the 12 months preceding the census. Individuals were also asked about history of fever during the preceding 30 days, and whether they had sought care at a health facility, community health worker, traditional healer or none. The specific health facility where the individual sought care was also specified to delineate health facility catchment areas according to the community. Finally, information was collected on the migration and mobility patterns of all participants, to better characterise the mobility profile of the district, and offer potential information on the sources of new infections if transmission was eventually significantly reduced.

Administrative and geographical information was obtained directly from the district authorities and from the most recent district profile of Magude published in 2014 by the central government authority[7] and district statistics performed by the National Institute of Statistics.[9 10] We retrospectively collected monthly data on inpatient discharges from the referral health centre of Magude Sede, as well as weekly outpatient malaria cases reported through the weekly epidemiological bulletin (Boletim Epidemiológico Semanal or 'BES' in Portuguese) for the period of 2010 to 2014. The BES did not distinguish between presumed and confirmed cases, the diagnostic used (rapid diagnostic test or microscopy) or the location of detection (health facility or in the community).

### Patient and public involvement

This study counted on the support and involvement of the community of Magude in all of its stages. During its design and planning, meetings were held with the district authorities and community leaders in order to inform them of the purpose of the DHP under the scope of the Magude project, as well as to landscape the number of villages in Magude that had to be covered by the DHP. The study team worked closely with village chiefs when performing the household visits, which ensured that all district households were covered by the DHP. Reports were submitted to the administrative authorities to support vector control activities, among others. Preliminary findings were also communicated in meetings held with the community prior to the deployment of MDAs, where the findings obtained from the study—particularly focussing on malaria—were discussed in detail.[11]

### Data analysis

Data management and descriptive analyses were performed using R Software.[12] The percentage distribution of categories within any given variable were calculated considering the observations with missing values in the denominator. No inferential statistics (p values or 95% CIs) were calculated

as the DHP covers the entire district, and therefore offers direct population parameters. The relationship between outpatient malaria cases and rainfall was evaluated using linear regressions for different rainfall lags. Spatial visualisation of the data collected on the field and obtained from web-based open source administrative data bases from DIVA-GIS[13] were performed using QGIS Software.[14]

Household's socio-economic status was measured using a modified version of Oxford's Poverty and Human Development Initiative Multidimensional Poverty Index (MPI),[15] following the same methodology used by the INDEPTH group to estimate the poverty index of various HDSS sites throughout Africa.[16] In summary, this method categorises households as 'deprived' or 'not deprived' based on six deprivation indicators: (1) lack of electricity; (2) lack or sharing of an improved sanitation facility; (3) lack of access to improved drinking water source, or source only available at more than 30 min walk, round trip; (4) sand floors in main houses; (5) dung, wood or charcoal used for cooking fuel and (6) households that do not own a car nor truck and do not own more than one of the following: radio, television, telephone, bike, motorbike or refrigerator. A deprivation indicator is generated as lowest deprivation (0 to 2), moderate deprivation (3 to 4) or highest deprivation.[5 6 16]

## RESULTS

### The geography of Magude district

The district of Magude has an area of 6961 km$^2$ and is divided in five administrative posts, namely: Magude Sede, Motaze, Panjane, Mahele and Mapulanguene (figure 1A). The vegetation of Magude is dominated by open forests and savannahs hosting animals such as impalas, warthogs, lions, buffalos and elephants. There is one permanent river (Incomati), which flows through the south-western region of the district and constitutes the main source of water in the area, and three intermittent rivers dependent on rainfall, called Massintonto, Uanétze and Mazimuchopes.[7]

### Demographic and socio-economic characteristics

The baseline census of 2015 registered 10 965 households and 52 802 individuals, of whom 48 448 (91.8%) were residents and 4133 (7.8%) were non-residents. The district had a population density of 6.9 inhabitants per km$^2$, which varies by administrative post. The majority of the population (73%) lives in the Magude Sede administrative post (the capital of the district); 14.1% in Motaze, 5.9% in Panjane, 3.7% in Mapulanguene and 3.4% in Mahele (table 1). The age and sex structure of the population is dominated by children and young people, and a sharp reduction of adults, particularly among males (figure 1B).

The census update conducted between August 2016 and October of 2016 registered a population of 61 868 individuals. Of the population censused in 2015, 42 792

**Table 1** Summary of Magude population in 2015 and 2016

| | District level | | Magude Sede | | Motaze | | Panjane | | Mahele | | Mapulanguene | |
|---|---|---|---|---|---|---|---|---|---|---|---|---|
| | N | % | N | % | N | % | N | % | N | % | N | % |
| Households 2015 | 10 965 | | 8011 | 73.1 | 1471 | 13.4 | 627 | 5.7 | 377 | 3.4 | 479 | 4.4 |
| Households 2016 | 11 960 | | 8520 | 71.3 | 1441 | 12.1 | 645 | 5.4 | 803 | 6.7 | 547 | 4.6 |
| Population 2015 | 52 802 | | 38 534 | 73.0 | 7421 | 14.1 | 3122 | 5.9 | 1784 | 3.4 | 1934 | 3.7 |
| Residents | 48 448 | 91.8* | 35 346 | 73.0 | 6605 | 13.6 | 2930 | 6.0 | 1695 | 3.5 | 1868 | 3.9 |
| Non-residents | 4133 | 7.8* | 2970 | 71.9 | 816 | 19.7 | 191 | 4.6 | 89 | 2.2 | 66 | 1.6 |
| Unclassified | 221 | 0.4* | 218 | 99.5 | 0 | – | 1 | 0.5 | 0 | – | 0 | – |
| Population 2016 | 61 868 | | 44 203 | 71.4 | 8149 | 13.2 | 3576 | 5.8 | 3499 | 5.7 | 2404 | 3.8 |
| Residents | 56943† | 92.1* | 40 623 | 71.4 | 7275 | 12.8 | 3361 | 5.9 | 3329 | 5.8 | 2322 | 4.1 |
| Non-residents | 4535‡ | 7.3* | 3266 | 72.0 | 842 | 18.6 | 206 | 4.5 | 143 | 3.2 | 76 | 1.7 |
| Unclassified | 390§ | 0.6* | 314 | 80.9 | 32 | 8.2 | 9 | 2.3 | 27 | 7.0 | 6 | 1.5 |
| Demographic events 2015–2016 | 3078 | | 2204 | 71.6 | 390 | 12.7 | 186 | 6.0 | 168 | 5.5 | 129 | 4.2 |
| Births | 1687 | 54.8 | 1203 | 71.4 | 220 | 13.0 | 92 | 5.5 | 91 | 5.4 | 80 | 4.7 |
| Immigrations | 721 | 23.4 | 521 | 72.3 | 60 | 8.3 | 51 | 7.1 | 60 | 8.3 | 29 | 4.0 |
| Deaths | 670 | 21.8 | 480 | 71.6 | 110 | 16.4 | 43 | 6.4 | 17 | 2.5 | 20 | 3.0 |

*Column %.
†56 943 = 48 448 (censused in 2015) – 616 (deaths) + 1670 (births) + 1235 (immigrations) + 6206 (censused in 2016).
‡4535=4133 – 42 (deaths) + 12 (births) + 64 (immigrations) + 368 (censused in 2016).
§390 = 221 – 8 (deaths) + 5 (births) + 21 (immigrations) + 151 (censused in 2016).

**Table 2** Socio-demographic characteristics of heads of households, and household characteristics (2015)

| Characteristics of heads of households* | | N | % |
|---|---|---|---|
| Sex | Males | 5051 | 54.6 |
| | Females | 4206 | 45.4 |
| Age group (years) | <18 | 40 | 0.4 |
| | 18–24 | 520 | 5.6 |
| | 25–64 | 6919 | 74.7 |
| | >=65 | 1777 | 19.2 |
| Education level | No formal education | 5460 | 59.0 |
| | 5th to 7th grade | 2611 | 28.2 |
| | 8th to 12th grade | 807 | 8.7 |
| | University | 37 | 0.4 |
| | Missing information | 342 | 3.7 |
| Marital status | Single | 4007 | 43.3 |
| | Married or de facto union | 3275 | 35.4 |
| | Divorced | 16 | 0.2 |
| | Separated | 393 | 4.3 |
| | Widow | 1554 | 16.8 |
| Residents of Magude | | 9257 | 84.4 |
| Households characteristics | | | |
| Average household size† | | 5 (3 to 7) | |
| Lone-resident households | | 1171 | 10.7 |
| Wall material of the main building | Cane | 3506 | 32.5 |
| | Cement | 2807 | 26.0 |
| | Mud bricks | 2338 | 21.6 |
| | Adobe | 1703 | 15.8 |
| | Zinc plates | 319 | 3.0 |
| | Wood | 128 | 1.2 |
| Type of toilet | Traditional latrine | 5774 | 53.5 |
| | Improved latrine | 1144 | 10.6 |
| | WC connected to septic tank | 233 | 2.2 |
| | No latrine | 3650 | 33.8 |
| Main source of lighting energy | Paraffin | 5362 | 49.6 |
| | Electricity | 3321 | 30.8 |
| | Candles | 1074 | 9.9 |
| | Solar panels | 502 | 4.6 |
| | Other | 539 | 5.0 |
| Kitchen Fuel | Wood logs | 6401 | 59.3 |
| | Coal | 740 | 6.9 |
| | Gas | 46 | 0.4 |
| | Electricity | 65 | 0.6 |
| | Paraffin | 8 | 0.1 |
| | Missing information | 3541 | 32,8 |
| Primary source of drinking water | Water pumps | 3700 | 34.3 |
| | Directly from the river | 2691 | 24.9 |
| | Piped water | 2218 | 20.5 |
| | Open well near the river | 1138 | 10.5 |
| | Other | 1054 | 9.8 |

Continued

| Table 2 | Continued | | | |
|---|---|---|---|---|
| **Characteristics of heads of households*** | | | **N** | **%** |
| Vector control tools | | | | |
| % of households with >1 ITN | | | 8906 | 81.2 |
| Universal ITN coverage (one net per two household members) | | | 5690 | 53.2 |
| IRS in past 12 months | | Yes | 5722 | 52.2 |
| | | No | 4719 | 43.0 |
| | | Unknown | 524 | 4.8 |

*or subhead (if head does not live in the household).
†Median (IQR).
IRS, indoor residual spraying; ITN, insecticide-treated net; WC, water closet.

(81%) were found during the update round, 6099 (11.5%) were reported to still be in Magude by a family member although fieldworkers did not find them in other households and therefore were not able to confirm and complete the registration of migration. There were an additional 3244 (6.1%) individuals censused in 2015 who were not found in 2016 and for whom there were no informants, but were still considered to be in Magude for this analysis. This update also recorded the death of 670 individuals censused in 2015, 1687 live births and 721 immigrations since the baseline census. An additional 7325 individuals who were missed in 2015 were censused during this update (table 1).

The baseline census indicated that 62% of males and 63.1% of females above the age of 14 reported being married or in de facto union; and 3% of married men practice polygamy. The prevalence of lack of formal education among those aged 6 and older in Magude is 51.9% among females and 47.4% among males. Approximately 37.2% of individuals reached fifth to ninth grade, and 9.5% have completed between tenth and twelfth grade (online supplementary table 1). In 2015 Magude had 31 primary schools offering first to fifth grades, 33 offering first to seventh grades, one secondary school offering eighth to twelfth grades and one private higher-education training centre. According to the national census performed in 2007, Xichangana is the main local language of the district and is the mother tongue of 92% of the population of Magude. However, 51% of the population also reports being able to speak Portuguese (the official language in Mozambique), which is the mother tongue of only 3.2% in the district.[7]

Occupations are unevenly distributed between males and females older than 18 years of age. The majority of the population (26.1% of males and 70.7% of females) relies on subsistence agriculture, fishing or working as cane cutters in the sugar plantations within Magude, or in Xinavane, in the nearby district of Manhiça. Other occupations include being a salesperson (8.9% of men and 11% of women), doing construction work (22.1% of men and 0.7% of women) or making coal (11.5% of men and 3.1% of women). A small proportion of the population works as security guards, or work in the public sector, particularly as health professionals, teachers or in the army (online supplementary table 1).

Men represent 54.6% of heads of households in Magude. Only 0.4% of household heads are younger than 18 years of age and 5.6% are between the ages of 18 to 24, while 74.7% are between 25 and 64 years old and 19.2% are older than 65 years of age (table 2). Forty-three per cent of heads of household are single, 35.4% are married or in de facto union, 16.8% are widows and 4.3% separated. Fifty-nine per cent of heads of households reported having no formal education, while 28.2% reported completing up to fifth to ninth grade, 8.7% up to tenth to twelfth grade and 0.4% reported having a university degree (table 2).

### Living conditions in Magude's households

Settlements in Magude are made of individual household compounds that are built in proximity to each other in the central areas of each administrative post, and more spread out in the rest of the district. The median household size is five, although 1171 households (10.7%) have only one resident, 40% of whom are older than 65 years old.

The majority of households are traditional round-shaped or rectangular-shaped huts constructed using cane (32.5%), cement (26%), mud bricks (21.6%) or reeds covered by adobe (15.6%) (table 2). More than half of the households have traditional latrines (53.5%), 10.6% have improved latrines, while 33.8% of households do not have any form of sanitation facility. Flush toilets can only be found in 2.2% of the households. The primary lighting sources used in the households are paraffin (49.6%), electricity (30.8%), candles (9.9%) and solar panels (4.6%). The majority of households cook using wood logs (59.3%). Water is mainly obtained from pumps (34.3%) or piped water (20.5%), although a quarter of the population collects water directly from the river (24.9%) or from open wells near the river (10.5%) (table 2).

With regards to household assets, 68.5% of households own at least one mobile phone, 36.6% own a radio, 32.7%

have a television, while 16.8% and 12.2% of households have a fridge and a freezer, respectively. Almost a quarter of households reported having bicycles (23.7%), while 10.8% reported owning cars, 5.3% motorbikes and 0.7% trucks (online supplementary figure 1).

Households in Magude had a median number of four deprivations. Twenty-five per cent of the households were in the lowest deprivation category, with 0 to 2 deprivation indicators; while 55.3% fell in the 3 to 4 deprivation group and 22.4% in the 5 to 6 deprivation group. This distribution was unevenly observed among the different administrative posts. Magude Sede had the highest proportion of households in the lowest deprivation group (32%), while Mapulanguene had the highest proportion of households in the moderate deprivation category (70.8%). Panjane and Mahele were the administrative posts with the highest proportion of highly deprived households (online supplementary table 2).

### Mobility patterns

The baseline census showed that 5% of residents reported having spent the night before the census outside of Magude (6.4% of males and 5% of females). Of these, 43% were younger than 14 years old, 25.1% were 15 to 29 years old, 18.1% were 30 to 44 years old, 10.3% were 45 to 64 years old and 3.4% were older than 65 years old (online supplementary table 3).

Almost half of the population (43% of males and 46.9% of females) did not report frequently travelling outside of Magude during the baseline census. Among the 54.4% who did, the places most frequently visited included Maputo City (55.3% of males and 58.3% of females), South Africa (23.1% of males and 16.8% of females), Gaza province (12.2% of males and 15.9% of females), Inhambane province (4.3% males and 2.6% females) and other Northern provinces within Mozambique (5.2% of males and 6.3% of females). Younger individuals (less than 15 years old) reported travelling the most to all provinces within the country, followed by those between the ages of 15 to 30; while 15 to 45 year olds are more likely to travel to South Africa. Individuals older than 45 reported travelling more to provinces northern of Maputo province, as well as abroad (online supplementary figure 2).

### Health and malaria information

The district has 10 functional health facilities (HFs)—one with an inpatient and maternity ward (Type I) and nine HFs only with maternity wards (Type II). Eight of the 10 HFs were active in 2015, and 2 more Type II HFs were added in 2016 and 2017. The referral HF, located in the Magude Sede administrative post, has 57 beds and three active medical doctors (0.06 doctors per 1000 inhabitants). The other nine Type II health facilities have 37 beds in total. The whole district has only 3 medical doctors, 15 general nurses and 19 maternal and child health nurses (figure 1A). Only three HFs have access to piped water, four have access to public network electricity and five rely only on solar panels. All health facilities are located on a local main road to facilitate access, and the median Euclidean distance from households to the nearest health facility is 2.7 km (IQR 1.4 to 7.9 km), although households are as close as 15 m or as far as 38.8 km. There is only one ambulance in Magude, which is used for transferring patients within the district, but also to the hospitals in Xinavane, Manhiça and Maputo.

The health system in Mozambique includes Community Health Workers (or APEs, from its acronym in Portuguese) who are trained by the Ministry of Health to offer primary health services in areas with poor access to HFs. They provide diagnosis and treatment for malaria, diarrhoea and pneumonia, and to refer patients with signs of sickness requiring higher medical attention.[17] In Magude there are 27 APEs distributed throughout the district (figure 1A). The average distance between households and the nearest APE is approximately 6.3 km (median of 4.4 km, IQR 2.8 to 5.4 km).

All HFs and APEs in Magude are equipped to diagnose malaria through Rapid Diagnostic Tests (RDTs) and light microscopy is only available in the Magude Sede Type I HF. They also offer treatment to all positive cases with Artemether-Lumefantrine, the first line treatment in Mozambique. Intermittent preventive treatment in pregnancy using sulfadoxine-pyrimethamine is also offered in all HFs of the district. In May of 2014 the NMCP conducted an LLIN universal distribution campaign, with 35 432 bed nets distributed in Magude district. This was followed by a focal IRS campaign between October 2014 and December 2014 using the insecticides deltamethrin and dichlorodiphenyltrichloroethane (DDT), which was only deployed in the Motaze administrative post, where the malaria burden was highest.

Malaria has traditionally been the first cause of disease in the district, responsible for approximately 53% of all consultations reported in 2003.[7] According to the inpatient discharge data available from January 2010 to December 2014, between 20% to 40% of all-age discharges in the inpatient department were due to malaria, followed by HIV (12% to 22%), anaemia (12% to 15%), pneumonia (6% to 12%) and diarrhoea (3% to 8%), varying by month and year (figure 2A). The weekly number of malaria cases reported through the BES in Magude between 2010 and 2014 follows a seasonal pattern with a peak between December and May and a reduction of cases during the dry season (figure 2B). According to BES data, there were 293 cases per 1000 population reported during the transmission year of July 2011 to June 2012, 247 per 1000 between July 2012 and June 2013, 252 cases per 1000 between July 2013 and June 2014 and 110 cases per 1000 between July 2014 and June 2015.[7] There is a fairly linear association between the rainfall in the preceding 2 months, and the number of malaria cases in Magude (linear regression coefficient=13.6, Rho=0.64, p value <0.001).

The prevalence of history of fever during the 30 days prior to the baseline census was 12.1% in children under the age of 5, 7.7% among 5 to 14 year olds and 11.6%

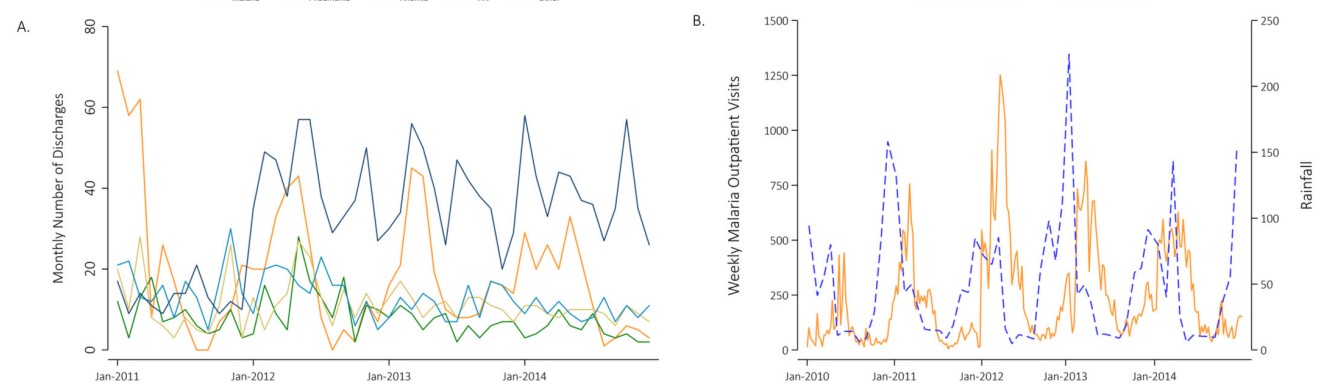

**Figure 2** Health profile of Magude district prior to the baseline census (2010 to 2014). (A) Most common diseases leading to all-age hospitalisation and later discharge in Magude reported by the District Health Authorities between 2010 and 2014. (B) Weekly number of outpatient malaria cases observed in Magude between 2010 and 2014 reported through the weekly epidemiological bulletin (BES) and monthly rainfall data obtained from the Climate Hazards Group InfraRed Precipitation with Station data (CHIRPS). BES, Boletim Epidemiológico Semanal.

in individuals older than 15 years. Almost all individuals reported going to a health facility as their primary source of healthcare (>99%), while a small proportion mentioned seeking care first from traditional healers or community health workers. The proportion of individuals who reported sleeping under a bed net the night before was 27.1% among children under the age of 5, 21.8% in 5 to 14 year olds and 27% among those >15 years of age (table 3). The universal bed net coverage (ie, one net for every two individuals of a household) in 2015 was 52.7% in the administrative post of Magude Sede, 59.9% in Motaze, 44.1% in Panjane, 52.7% in Mahele and 52.1% in Mapulanguene. Finally, the reported coverage of IRS during the previous 12 months was 49% in Magude Sede, 79% in Motaze, 41.1% in Panjane, 61% in Mahele and 30.7% in Mapulanguene (table 2 and online supplementary table 4).

## DISCUSSION

Overall, the Demographic and Health Platform established in Magude offered detailed insight into the baseline socio-demographic profile of Magude district prior to the Magude project. It revealed that the number of residents in Magude in 2015 (52 804 individuals and 10 965 households) was similar to the number reported by the 2007 national census conducted by the National Institute of Statistics (INE) (53 229 individuals and 11 408 households)[9] but did not coincide with the projected population for Magude in 2015 estimated by the INE to be 62 000,[10] which is not surprising acknowledging the limitations of population projections. The census update conducted in 2016 identified individuals who had been missed by the baseline census, and recorded the births, deaths and in-migrations since baseline. As a result, the population of Magude in 2016 assuming that

| Table 3 Individual-level health and malaria prevention indicators in Magude district 2015 | | | | | | |
|---|---|---|---|---|---|---|
| | **<5** | | **5 to 14** | | **>15** | |
| **Age group in years** | **N** | **%** | **N** | **%** | **N** | **%** |
| Had fever in the preceding 30 days | | | | | | |
| Yes | 1043 | 12.1 | 1128 | 7.7 | 2931 | 11.6 |
| No | 7492 | 87.2 | 13 325 | 91.2 | 21 977 | 87.1 |
| Unknown | 55 | 0.6 | 154 | 1.1 | 319 | 1.3 |
| Primary source of healthcare | | | | | | |
| Health facility | 8552 | 99.6 | 14 548 | 99.6 | 25 075 | 99.4 |
| Traditional healer | 13 | 0.2 | 27 | 0.2 | 61 | 0.2 |
| Community health worker | 22 | 0.3 | 32 | 0.2 | 66 | 0.3 |
| Slept under a bed net the preceding night | | | | | | |
| Yes | 2330 | 27.1 | 3179 | 21.8 | 6799 | 27.0 |
| No | 6260 | 72.9 | 11 428 | 78.2 | 18 428 | 73.0 |

those censused in 2015 who were not found in 2016 were still living in the district, was 61 868 individuals, which is similar to the projection by INE for this year (62 924).

The usefulness of a demographic and health surveillance system in demographic and biomedical research in African countries has been previously described.[18] Having a precise source of population data is crucial for the correct estimation of the risk of disease and coverage of health indicators in a population, as well as for planning purposes of public health interventions.[19] This platform has shown that detailed population data collected at individual level at one point in time still misses a number of individuals, who can be captured after a later update. However, all the census rounds faced the challenge of finding individuals at home, or informants of unavailable individuals, to identify whether a member lives in the area despite not being found during a specific census round. To identify or collect information about absent individuals, field supervisors attempt to arrange interviews with them at their places of work, or interview their family members at their households during weekends. The same phenomenon is expected to take place when implementing malaria interventions in the community (such as rounds of IRS or of mass drug administrations), which generally challenges the estimation of intervention coverage.

Household-level data indicates that the households in Magude are typical of a rural area with suboptimal conditions regarding access to water, electricity and sanitation. The majority of houses were relatively homogeneous, built with non-durable materials, such as cane and adobe and did not have access to clean water, conditions indicative of a low socio-economic status.[20] The age and sex composition of Magude's population is typical of rural Mozambique and sub-Saharan Africa, composed primarily by children and young individuals and a decreasing proportion of adults as age increases.[21] Another important aspect of the structure of the population of Magude is the lower sex ratio (number of males in relation to that of females) among young and middle-aged adults, which has been reported, in the neighbouring district of Manhiça, to be related to male labour migration to Maputo city or South Africa and higher male mortality.[21]

A large proportion of the heads of households and of the overall population of Magude reported not having received a formal education, a major risk factor for health outcomes. Additionally, most individuals reported having occupations related to agriculture or fishing, which are usually carried out early in the morning until around noon. These aspects should be taken into consideration when planning mobilisation campaigns and community-based interventions, as visits might have to be conducted at certain times of day and/or venues outside the households to be able to find these individuals.

Magude is also subject to significant mobility and migration, as more than half of the population reported travelling frequently outside of the district. The destinations most reported by travellers living in Magude were Maputo and South Africa, followed by Gaza and Inhambane provinces. This information is useful to evaluate the risk of malaria importation from other areas due to travel or migration.[22] Maputo city and South Africa are areas with lower malaria burden than Magude;[23] however, the district of Manhiça surrounding Magude, the provinces of Gaza, Inhambane and the rest of Mozambique have higher malaria prevalence estimates than Magude district,[2] and could be a source of imported infections.[24] These places are commonly visited by all age groups, whereas the lower-endemic areas are mainly visited by <30 year olds. This age pattern of migration is typical of the rural areas of southern Mozambique, where migration rates are higher among individuals aged 20 to 45 years and their children, and decrease with increasing age when individuals establish as permanent residents in an area.

The population of Magude has access to 10 health facilities, one of them with an inpatient ward with 57 beds. Thus there are 11.8 beds per 10 000 population, a ratio that is higher than the national level (7 beds per 10 000 in Mozambique).[25] While most respondents indicated seeking healthcare primarily from formal health facilities, the majority of the population is scattered throughout the district as far as 38 kms away from a HF and it is difficult to access health facilities or APEs due to limited roads and lack of transportation. This has implications for the delivery of public health interventions, especially for those that rely on the passive detection of cases to control transmission in the community.

Data between 2010 and 2014 from the outpatient and inpatient department of the Magude Sede health centre indicated that malaria was the main cause of hospitalisation and later discharged with a clear seasonal pattern. The population of Magude is also burdened by HIV, anaemia, pneumonia and diarrhoea, similar to its neighbouring districts of Manhiça and Chokwé.[26–33] Inpatient information was retrospectively collected, and thus, its quality and accuracy depends on the data at the time at which it was collected. It may also be biassed by disease-specific interventions or changes in diagnostics, referral protocols of severe cases to other districts or reporting practices. Additionally, cases reported by BES are difficult to interpret given their lack of disaggregation in presumed or confirmed, diagnostic tool used or place of detection of the case. Thus, they are subject to changes in RDT use and stock-outs, in care seeking behaviour to HFs or APEs, or reporting inaccuracies. In fact, cases reported by the APEs are usually excluded by those who report in BES, and neither the total number of fevers nor the number of individuals tested are reported. Thus, a stronger surveillance system than BES was thought to be required, and consequently established in Magude in 2015, to fully capture the clinical malaria profile of a district aiming to eliminate malaria.[4 34]

The age-specific self-reported history of fever in the past 30 days (7.7% to 12.1%) were slightly lower than the average percentage of fevers reported in the previous 2

weeks in Maputo province, which was 15% according to the most recent malaria indicator survey.[2] Approximately half of the households in Magude reported owning one bed net for every two individuals (universal coverage). However, only a fourth of the population reported sleeping under a bed net the previous night. These findings mirror the estimates reported for Maputo province in 2011 by the Demographic Health Survey, which found a reported bed net usage rate of 27% in Maputo province.[35] This information is indicative of an area where bed net owners do not necessarily use the net as a preventive tool against malaria during the rainy season, despite this being a prevalent disease in the district. Other demographic and socio-economic risk factors that have been associated with LLIN use, such as households with more children, household heads with no formal education or households far away from the health facilities, might explain the low bed net usage in Magude.[36–38]

The reported IRS coverage in the area of Motaze was higher, which corresponds with the focal IRS campaign conducted by the NMCP that took place only there in September of 2014. The coverage reported in other areas was probably the consequence of recall bias, as a districtwide IRS campaign took place in 2013. Overall, the coverage reported for LLINs and IRS throughout the district was below the WHO recommended coverage of >80%,[39] leaving a significant proportion of the population unprotected by any of the standard preventative measures. The identification of gaps in such vector control coverages called for a strong community engagement campaign focussed on the use of the bed nets and participation in the yearly rounds of IRS.

## CONCLUSION

Through the establishment of a Demographic and Health Platform in Magude in 2015, which was updated in 2016, and a baseline assessment of the relevant health indicators, it was possible to fully characterise a district where malaria elimination interventions were to be deployed and evaluated. Magude represents a typical rural district of Mozambique characterised by limited social and economic infrastructures, which had to be considered for the design and operationalisation of the community-based interventions. This study also revealed a low education level among a large proportion of the population and identified agriculture as the main economic activity in the district. These socio-demographic characteristics suggest that a strong community engagement would be necessary for the implementation of a malaria elimination project. Half of the population of Magude reported travelling outside of the district to areas of high and low malaria transmission intensity, showing that malaria importation will likely be a source of continuous transmission even if interruption of local transmission is achieved. Also, the poor access to healthcare services, as well as to core malaria vector control interventions such as IRS and bed nets distributed by the NMCP, indicated that these

aspects needed to be integrated within the malaria elimination programme planned for the district, in order to maximise the impact of the interventions aiming to interrupt transmission. Overall, this district represents the reality of the majority of malaria endemic areas in sub-Saharan Africa, where elimination is most needed, and where it is most pressing to demonstrate the feasibility of elimination strategies.

**Author affiliations**
[1]Centro de Investigação em Saúde da Manhiça (CISM), Manhiça, Mozambique
[2]Barcelona Institute for Global Health (ISGlobal), Hospital Clínic-Universitat de Barcelona, Barcelona, Spain
[3]National Directorate of Health, Ministry of Health, Maputo, Mozambique
[4]ICREA, Pg. Lluís Companys 23, 08010, Barcelona, Spain
[5]Pediatric Infectious Diseases Unit, Pediatrics Department, Hospital Sant Joan de Déu (University of Barcelona), Barcelona, Spain
[6]Harvard T.H. Chan School of Public Health, Boston, MA, United States
[7]National Institute of Health, Ministry of Health, Maputo, Mozambique

**Acknowledgements** We would like to thank the community of Magude for participating in this study, and the team of field workers who have collected the data presented here. We would also like to acknowledge the Administrative and District Health Authorities of Magude for their collaboration and for providing some of the information also included in this article. We thank everyone who supported this study directly or indirectly through fieldwork or analysis support.

**Contributors** BG: participated in the study design and fieldwork, supported in the data cleaning and data analysis process and wrote the draft of this article. AN: participated in the study design, in study analyses, in the interpretation of results and writing of this article. HM: cleaned and analysed the data, and contributed to the writing of this article. HM: led the implementation of field activities and data collection process, and participated in the interpretation of results. EJ: participated in the study design and implementation of field activities and data collection. CG: participated in the interpretation of results and writing of this article. LC: participated in the study design, interpretation of results and writing of this article. FA: designed the data collection tools, supported the implementation of field activities and data cleaning. EM: participated in the study design, and interpretation of results. QB: participated in the study design, interpretation of results and writing of this article. RR: participated in the interpretation of results and writing of this article. PA: participated in the study design, interpretation of results and writing of this article. PA: participated in the study design, field implementation interpretation of results and writing of this article. FS: participated in the study design, supervised field activities, interpretation of results and writing of this article. CS: conceived the study, participated in interpretation of results and writing of this article.

**Funding** Funding was provided by the Bill & Melinda Gates Foundation and the Fundación 'la Caixa' Partnership for the Elimination of Malaria in Southern Mozambique (OPP1115265). QB is an ICREA (Institut Catal. de la Recerca i Estudis Avan.ats; Catalan Government) Research Professor. ISGlobal is a member of the CERCA Programme, Generalitat de Catalunya.

**Map disclaimer** The depiction of boundaries on the map(s) in this article do not imply the expression of any opinion whatsoever on the part of BMJ (or any member of its group) concerning the legal status of any country, territory, jurisdiction or area or of its authorities. The map(s) are provided without any warranty of any kind, either express or implied.

**Competing interests** None declared.

**Patient and public involvement** Patients and/or the public were involved in the design, or conduct, or reporting or dissemination plans of this research. Refer to the Methods section for further details.

**Patient consent for publication** Not required.

**Ethics approval** The Demographic and Health Platform protocol, consent forms and questionnaires were approved by the CISM internal ethics committee and written consent from the health authorities was also sought prior to its implementation. Meetings were held with community leaders and with general members of the community of Magude to inform them about the DHP operations and the malaria elimination project as a whole. A written informed consent was

obtained from all household heads to record household-level information, as well as from all individuals providing individual information. Informed consents for children under the age of 18 years were sought from their parents or primary caretakers. The collection of district health surveillance data was conducted under the protocol that aimed to evaluate the impact of the Magude project on malaria transmission, which was approved by CISM's internal ethical committee, the Ethics Committee of the Hospital Clínic of Barcelona and the Ministry of Health National Bioethics Committee of Mozambique (IRB00002657).

**Provenance and peer review** Not commissioned; externally peer reviewed.

**Data availability statement** Data are available upon reasonable request. Data may be obtained from a third party and are not publicly available. The data sets collected as part of the Demographic Health Platform and analysed during the current study are available from the corresponding author upon written request. The data presented in Figures 2a and 2b belongs to the Ministry of Health of Mozambique and is considered as third-party data which can be accessed by contacting the following individuals appointed by the Ministry of health: To gain access to Inpatient Department data collected by the Ministry of health please contact Dr Baltazar Candrinho, Head of NMCP Mozambique, phone number: +258828665730 or e-mail: candrinhobaltazar@gmail.com. To gain access to the data collected through the Boletim Epidemiologico Semanal (BES) please contact Dr Lorna Gujral, Department of Epidemiology/MOH/Mozambique, phone number: +258823250800 or e-mail: lornita15.lgg@gmail.com

**ORCID iDs**
Beatriz Galatas http://orcid.org/0000-0002-9546-6385
Ariel Nhacolo http://orcid.org/0000-0003-3206-2372

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
