## [Reviewer comments · BMJ Open]

ARTICLE DETAILS

TITLE (PROVISIONAL)	Demographic and health community-based surveys to inform a malaria elimination project in Magude district, Southern Mozambique
AUTHORS	Galatas, Beatriz; Nhacolo, Ariel; Marti, Helena; Munguambe, Humberto; Jamise, Edgar; Guinovart, Caterina; Cirera, Laia; Amone, Felimone; Macete, Eusebio; Bassat, Quique; Rabinovich, Regina; Alonso, Pedro; Aide, Pedro; Saute, Francisco; Sacoor, Charfudin

VERSION 1 – REVIEW

REVIEWER	David Larsen Syracuse University, USA
REVIEW RETURNED	17-Sep-2019

GENERAL COMMENTS	Major comments  1. I would like to see a breakdown of poverty using the multi-dimensional poverty index. It appears that the data are at least mostly there. 2. Throughout the authors need to specify whether the malaria cases are laboratory confirmed (and what method), and the location of presentation (such as health facility). 3. On Table 2, the education categories don't make too much sense. The categories jump from illiterate to 5-7th grade. I would separate literacy from educational attainment (grade level). Also the education is labeled as the level for kids six and older but the table is for heads of household. Also the marital status is limited to > 14 years old, but these are heads of household so probably don't need that. And lastly perhaps move the total residents to the bottom of the head of household section of the table. 4. On table 2, vector control tools I'd like to see percentage of households with at least one ITN. Minor comments  5. Perhaps include "census" and "demographic surveillance site" in key words. 6. In the abstract line 48 can you clarify what the project is and what the project's interventions are? Or just delete that last bit of the sentence? 7. Intro, page 6 lines 26-27. Can you specify what metric the >50% and 3% of malaria transmission is, is that parasite prevalence, child parasite prevalence or something else? 8. Page 7, line 83 – are these laboratory-confirmed malaria cases? And RDT or microscopy?
---

REVIEWER	Jean Gaudart Aix Marseille University, France
-----------------	--

REVIEW RETURNED	21-Jan-2020
-------------

GENERAL COMMENTS	In this very interesting paper, the authors aimed at describing the Magude district before the implementation of malaria control program, developing a demographic health platform. This is a very good idea, as this kind of description is of high importance before developing new control strategies. But I've some questions in order to clarify the article, mainly about the analysis itself. The authors stated that no inferential statistics are needed. I agree with this approach if, and only if, the authors are absolutely sure to have exhaustive data. In my opinion, it is not certain to have, e.g., all malaria cases reported. But, nevertheless, even if authors have chosen not to provide inferences, it will be very helpful to use classical methods providing synthetic statistics. Among them: hierarchical classification, principal component analysis, multiple correspondence analysis. Indeed, the currently result paragraph appears to be a list of characteristics, and these statistical (non-inferential) methods are very important to highlight the main characteristics. Furthermore, I have understood that the reported malaria cases were not related to the demographic and sociological informations. Why? In a health and demographic platform, with an ID for each household, it should be possible to link each case to the correspondent information. This will be very important in order to highlight the main demographic and sociologic risk factors and to map malaria cases. And if it is not possible in which aspect this platform is a "health" one (and not only a demographic platform)? I know that, on the field, it is not always feasible. But, at least, in my humble point of view, it is important to analyze the malaria time series, together with the relationships with meteorological and environmental factors. At least rainfall, temperature, vegetation, data (or at least estimations) are freely available on the internet. Another aspect that could be explored is the relationships between the demographic and sociological factors and the experience of malaria as both informations are present in the questionnaire. Minor: The number of health facility is not clear: 8, 9 or 10?
--

VERSION 1 – AUTHOR RESPONSE

Reviewer: 1

Major comments

1. I would like to see a breakdown of poverty using the multi-dimensional poverty index. It appears that the data are at least mostly there.

Authors' response: We thank the reviewer for the suggestion and have proceeded to compute a modified version of the multidimensional poverty index. To do this, we have followed the same methodology used by the INDEPTH group to estimate the poverty index of various HDSS sites throughout Africa, in which Manhiça district (neighboring Magude, and district where the Manhiça Health Research Center is located). Please refer to this article for further details on the modified poverty index used (Matthew M. Coates, Mamusu Kamanda, Alexander Kintu, Iwara Arikpo, Alberto

Chauque, Melkamu Merid Mengesha, Alison J. Price, Peter Sifuna, Marylene Wamukoya, Charfudin N. Sacoor, Sheila Ogwang, Nega Assefa, Amelia C. Crampin, Eusebio V. Macete, Catherine Kyobutungi, Martin M. Meremikwu, Walter Otieno, Kafui Adjaye-Gbewonyo, Andrew Marx, Peter Byass, Osman Sankoh & Gene Bukhman (2019) A comparison of all-cause and causespecific mortality by household socioeconomic status across seven INDEPTH network health and demographic surveillance systems in sub-Saharan Africa, *Global Health Action*, 12:1, 1608013, DOI: 10.1080/16549716.2019.1608013). We have added the following paragraphs to the methods and results section:

Methods (Lines 208-216): “Household’s socio-economic status was measured using a modified version of the Multidimensional Poverty Index (MPI) developed by the Oxford Poverty and Human Development Initiative [15], following the same methodology used by the INDEPTH group to estimate the poverty index of various HDSS sites throughout Africa [16]. In summary, this method categorizes households as “deprived” or “not deprived” based on six deprivation indicators: 1) lack of electricity, 2) lack or sharing of an improved sanitation facility; 3) lack of access to improved drinking water source, or source only available more than 30-minute walk, round trip; 4) sand floors in main houses; 5) dung, wood or charcoal used for cooking fuel; and 6) households that do not own a car nor truck and do not own more than one of the following: radio, TV, telephone, bike, motorbike, or refrigerator. A deprivation indicator is generated as lowest deprivation (0-2), moderate deprivation (3-4), or highest deprivation (5-6) [16] “

Results (lines 286-292): “Households in Magude had a median number of 4 deprivations. Twenty-five percent of the households were in the lowest deprivation category, with 0-2 deprivation indicators; while 55.3% fell in the 3-4 deprivation group and 22.4% in the 5-6 deprivation group. This distribution was unevenly observed among the different administrative posts. Magude Sede had the highest proportion of households in the lowest deprivation group (32%), while Mapulanguene had the highest proportion of households in the moderate deprivation category (70.8%). Panjane and Mahele were the administrative posts with the highest proportion of highly deprived households (Sup. Table 2).“

2. Throughout the authors need to specify whether the malaria cases are laboratory confirmed (and what method), and the location of presentation (such as health facility).

Authors’ response: We understand the reviewer’s petition. Unfortunately, the data collected through weekly epidemiological bulletin (Boletim Epidemiológico Semanal or “BES” in Portuguese) did not distinguish between presumed and confirmed cases, cases confirmed by RDT or microscopy or the location of detection (at the HF or the community). We have explained this in the methods section (lines 184-191). We also identified this as a limitation in the “Strengths and limitations” section and in the discussion (lines 417-429). In fact, this was identified as one of the major limitations to evaluate the malaria elimination project, and a strengthened weekly surveillance system was established in 2015 by our team as a result.

3. On Table 2:

a) the education categories don’t make too much sense. The categories jump from illiterate to 5-7th grade. I would separate literacy from educational attainment (grade level). Also the education is labeled as the level for kids six and older but the table is for heads of household. Also the marital status is limited to > 14 years old, but these are heads of household so probably don’t need that

Authors’ response: We thank the reviewer for pointing these mistakes out. We have corrected the education category to 5th to 9th grade. For consistency purposes, we have substituted the category “Illiterate” to “No formal education”, so that we respect the way the question was formed. Finally, we

have removed the age ranges from the “Education” and “Occupation” category in order to avoid confusion.

b) And lastly perhaps move the total residents to the bottom of the head of household section of the table.

Authors’ response: We have adapted the table as suggested.

4. On table 2, vector control tools I’d like to see percentage of households with at least one ITN.

Authors’ response: 81.2% (n=8906) households had at least one insecticide treated net (ITN) in 2015. We have added this information to Table 2.

Minor comments

5. Perhaps include “census” and “demographic surveillance site” in key words.

Authors’ response: Both words been added as a key word.

6. In the abstract line 48 can you clarify what the project is and what the project’s interventions are? Or just delete that last bit of the sentence?

Authors’ response: We have corrected the abstract to introduce the main objective of the Magude project (to evaluate the feasibility of malaria elimination in southern Mozambique). However, due to the word limit of the abstract it was not possible to include a full description of the project.

7. Intro, page 6 lines 26-27. Can you specify what metric the >50% and 3% of malaria transmission is, is that parasite prevalence, child parasite prevalence or something else?

Authors’ response: Thank you for pointing this out. We have specified that the prevalence estimates provided are for children of 6 to 59 months of age.

8. Page 7, line 83 – are these laboratory-confirmed malaria cases? And RDT or microscopy?

Authors’ response: As specified above, the data we used come from the public health facilities, compiled by the district health authorities and unfortunately they do not distinguish the data by such categories.

Reviewer: 2

Reviewer Name: Jean Gaudart.

Institution and Country: Aix Marseille University, France

Please state any competing interests or state ‘None declared’: None declared

• In this very interesting paper, the authors aimed at describing the Magude district before the implementation of malaria control program, developing a demographic health platform. This is a very good idea, as this kind of description is of high importance before developing new control strategies. But I’ve some questions in order to clarify the article, mainly about the analysis itself.

Authors’ response: We thank the reviewer for his appreciation of our study.

• The authors stated that no inferential statistics are needed. I agree with this approach if, and only if, the authors are absolutely sure to have exhaustive data. In my opinion, it is not certain to have, e.g., all malaria cases reported. But, nevertheless, even if authors have chosen not to provide inferences, it

will be very helpful to use classical methods providing synthetic statistics. Among them: hierarchical classification, principal component analysis, multiple correspondence analysis. Indeed, the currently result paragraph appears to be a list of characteristics, and these statistical (non-inferential) methods are very important to highlight the main characteristics.

Authors' response: We agree with this comment, which has also been raised by the first reviewer. As such, we have proceeded to calculate a modified version of the multi-dimensional poverty index following the same methodology that was used to estimate the same metric in all the INDEPTH HDSS sites in Africa. Please refer to our answer to comment #1 of the first reviewer for more details on the analysis conducted and the results obtained.

- Furthermore, I have understood that the reported malaria cases were not related to the demographic and sociological informations. Why? In a health and demographic platform, with an ID for each household, it should be possible to link each case to the correspondent information. This will be very important in order to highlight the main demographic and sociologic risk factors and to map malaria cases. And if it is not possible in which aspect this platform is a "health" one (and not only a demographic platform)?

Authors' response: The malaria case outpatient and inpatient data presented in this manuscript corresponds to the data collected retrospectively in an aggregated form prior to the establishment of the demographic and health platform in Magude. Therefore, it is not possible to link previous malaria cases (recorded in health facilities between 2010 and 2014) with the ID that was assigned to each individual in 2015. However, a strengthened surveillance system was established by our team in Magude from 2015 onwards as part of the malaria elimination project. With this reinforced surveillance system we were able to link clinical malaria cases to their households, and a response (reactive focal antimalarial drug administration) was triggered. Additionally, the data collected through the census served as a sampling frame for the selection of individuals in order to implement annual parasite surveys to measure parasite prevalence and to collect data on demographic, household, and community risk factors. The results of all of these activities will be presented in subsequent publications, but are outside of the scope of this particular manuscript that envisaged to present the district of Magude at baseline.

Regarding the reviewer's comments on the "health" component of the demographic and health platform – Despite not being able to retrospectively match the aggregated malaria cases to individual-level socio-demographic data, this platform allowed for the collection of baseline and follow-up data to monitor community-level health and malaria indicators. Thus, since 2017 (despite not being included in this manuscript), individual-level outpatient malaria case data were linked to the census database to trigger a response and to understand the factors associated with clinical malaria episodes. All this considered, we believe that it is appropriate to consider this platform a "demographic and health" platform.

- I know that, on the field, it is not always feasible. But, at least, in my humble point of view, it is important to analyze the malaria time series, together with the relationships with meteorological and environmental factors. At least rainfall, temperature, vegetation, data (or at least estimations) are freely available on the internet.

Authors' response: We agree with the reviewer. We have extracted rainfall information from the raster files from the Climate Hazards Group Infrared Precipitation with Station data (CHIRPS – please refer to this article for further information: <https://www.nature.com/articles/sdata201566>). The mean of the monthly rainfall has been added to figure 2B.

• Another aspect that could be explored is the relationships between the demographic and sociological factors and the experience of malaria as both informations are present in the questionnaire.

Authors' response: Please refer to our answer to the previous comment related to the ability to associate malaria cases with their socio-demographic characteristics.

Minor:

The number of health facility is not clear: 8, 9 or 10?

Authors' response: There are 10 health facilities in total. We agree with the reviewer that this was not clearly explained in the previous version of the manuscript and have corrected to clarify this information (lines 308-318).

VERSION 2 – REVIEW

REVIEWER	David Larsen Syracuse University Department of Public Health
REVIEW RETURNED	20-Feb-2020

GENERAL COMMENTS	The authors have outlined the development of a demographic and health program that will facilitate the evaluation of malaria elimination operations in southern Mozambique. The article is well written and the results are clearly presented.
--

REVIEWER	jean Gaudart Aix Marseille University, France
REVIEW RETURNED	27-Feb-2020

GENERAL COMMENTS	The authors have answered all my previous comments. I will be very happy if the author analyse the relationship between rainfall and malaria incidence, also estimating the time lag between the 2 curves. not only plotting a figure.
--

VERSION 2 – AUTHOR RESPONSE

Reviewer 2:

I will be very happy if the author analyse the relationship between rainfall and malaria incidence, also estimating the time lag between the 2 curves. not only plotting a figure.

Author's response: We have added this analysis to the methods and the results section. We hope this is enough for the reviewer. Any further analysis exploring case incidence and many covariates will be presented in more detailed in a separate publication.

Lines 209-210: ". The relationship between outpatient malaria cases and rainfall was evaluated using linear regressions for different rainfall lags" and Lines 351-353 "There is a fairly linear association between the rainfall in the preceding 2 months, and the number of malaria cases in Magude (linear regression coefficient=13.6, Rho=0.64, p-value<0.001).".